

# Asymmetric butterfly velocities in 2-local Hamiltonians

Yong-Liang Zhang[1] and Vedika Khemani[2]

**1** Department of Physics and Institute for Quantum Information and Matter,
California Institute of Technology, Pasadena, CA 91125, USA
**2** Department of Physics, Stanford University, Stanford, CA 94305, USA

## Abstract

The speed of information propagation is finite in quantum systems with local inter-actions. In many such systems, local operators spread ballistically in time and can be characterized by a "butterfly velocity", which can be measured via out-of-time-ordered correlation functions. In general, the butterfly velocity can depend asymmetrically on the direction of information propagation. In this work, we construct a family of simple 2-local Hamiltonians for understanding the asymmetric hydrodynamics of operator spreading. Our models live on a one dimensional lattice and exhibit asymmetric butterfly velocities between the left and right spatial directions. This asymmetry is transparently understood in a free (non-interacting) limit of our model Hamiltonians, where the butterfly speed can be understood in terms of quasiparticle velocities.


## 1 Introduction

Understanding the quantum dynamics of thermalization in isolated many-body systems is a topic of central interest. While memory of a system's initial conditions is always preserved

under unitary dynamics, this information can get "scrambled" and become inaccessible to local measurements, thereby enabling *local* subsystems to reach thermal equilibrium [1–4]. This scrambling can be quantified by studying the spatial spreading of initially local operators under Heisenberg time evolution. Under dynamics governed by a local time-independent Hamiltonian $H$, an initially local operator near the origin, $A_0$, evolves into $A_0(t) = e^{iHt}A_0 e^{-iHt}$. As $A_0(t)$ spreads in space, it starts to overlap with local operators $B_x$ at spatially separated locations $x$. The effect of scrambling is thus manifested in the non-commutation between $A_0(t)$ and $B_x$, which can be quantified via an out-of-time-ordered correlator (OTOC): [5–52]

$$C(x,t) = \Re\langle A_0^\dagger(t)B_x^\dagger A_0(t)B_x\rangle = 1 - \frac{1}{2}\langle [A_0(t),B_x]^\dagger [A_0(t),B_x]\rangle, \tag{1}$$

where $A_0, B_x$ are local unitary operators, $\Re$ represents the real part, and the expectation value $\langle\rangle$ is with respect to the infinite temperature thermal ensemble.

The OTOC is expected to exhibit the following features in systems with scrambling dynamics [6,7,11,17,19,24,53–67]: At early times, $A_0(t)$ approximately commutes with $B_x$ and the OTOC is nearly equal to one. At late times, $A_0(t)$ becomes highly non-local and spreads across the entire system, and the OTOC decays to zero [11,14,21,25]. At intermediate times, the operator has most of its support within a region around the origin defined by left and right operator "fronts" that propagate outwards, and generically also broaden in time [58,59]. As the operator front approaches and passes $x$, the OTOC $C(x,t)$ decays from nearly one to zero. We will restrict ourselves to translationally invariant systems where operators spreads ballistically with a butterfly speed $v_B$, which is similar in spirit to the Lieb-Robinson speed [68] characterizing the speed of information propagation. In these cases, the operator fronts define a "light-cone" within which the OTOC is nearly zero.

A set of recent papers illustrated that the butterfly velocity can depend on the direction of information spreading [69,70]. In one dimension, the asymmetry between the different directions can be quantified by the butterfly speeds $v_B^r$ and $v_B^l$, where the superscript $r$ ($l$) represent propagation directions to the right (left). While local unitary circuits can be 'chiral' and exhibit maximally asymmetric information transport (corresponding to one of $v_B^r$ or $v_B^l$ equal to zero), this chirality is 'anomalous' for time-independent Hamiltonians in one dimension [71]. Thus, the existence of asymmetric information spreading in Hamiltonian models was an open question, recently addressed by Refs. [69,70]. Ref. [70] constructed models of asymmetric (but not fully chiral) unitary circuits, and obtained Hamiltonians derived from such circuits that needed a minimum of three-spin interactions and were numerically shown to have asymmetric butterfly speeds. On the other hand, Ref. [69] showed how this asymmetry could be induced by anyonic particle statistics.

In this work, we present a complementary and physically transparent way for constructing a family of two-local Hamiltonians with asymmetric information propagation. Our construction does *not* rely on particle statistics, nor is it inspired by unitary circuits. Instead, we start with non-interacting integrable spin 1/2 models where the butterfly speed is related to quasiparticle propagation velocities and can be analytically calculated [66,67,72,73]. We show how the butterfly speed can be made asymmetric in such models, before generalizing to non-integrable Hamiltonians by adding interactions. The model and mechanism we present for obtaining asymmetric butterfly velocities is orthogonal to prior works on this topic, and provides a counterexample to the claim that one needs exotic anyonic particle statistics for asymmetric transport [69] — instead showing how this feature can be simply and generically obtained in solvable free fermionic models. Indeed, providing 'minimal models' for physical phenomena are often helpful in distilling necessary ingredients, and our work serves this purpose by furnishing much simpler classes of models with asymmetric information spreading than prior examples in the literature.

## 2 Integrable Hamiltonians

In this section, we construct time-independent integrable Hamiltonians for spin 1/2 degrees of freedom living on an infinite one dimensional lattice. The Hamiltonians only have local terms acting on 2 spins at a time. These models are exactly solvable, so the butterfly velocities can be analytically calculated, and demonstrated to be asymmetric. This family of Hamiltonians parameterized by $\lambda$ takes the form:

$$
H_\lambda = -\frac{J(1-\lambda)}{2} \sum_j \left[ h_{yz} \sigma_j^y \sigma_{j+1}^z + h_{zy} \sigma_j^z \sigma_{j+1}^y \right]
$$
$$
\frac{-J\lambda}{2} \sum_j \left[ h_{zz} \sigma_j^z \sigma_{j+1}^z + h_x \sigma_j^x \right], \tag{2}
$$

where $\sigma_j^x, \sigma_j^y, \sigma_j^z$ are the Pauli spin 1/2 operators located at site $j$, $J > 0, h_{zz}, h_x, h_{yz}, h_{zy}$ are constants, and the parameter $\lambda$ lies in the range $[0,1]$. This model can be mapped to a system of free fermions via a Jordan-Wigner representation. When $\lambda = 1$, the Hamiltonian is the well known transverse Ising model with inversion symmetry about the center of the chain. On the other hand, for $\lambda < 1$, the Hamiltonian does not have inversion symmetry when $h_{yz} \neq h_{zy}$.

In order to detect the ballistic light cone and asymmetric butterfly velocities, we consider the OTOCs

$$
C_{\mu\nu}(j,t) = \Re \langle \sigma_0^\mu(t) \sigma_j^\nu \sigma_0^\mu(t) \sigma_j^\nu \rangle_{\beta=0}, \tag{3}
$$

where $\mu, \nu \in \{x, y, z\}$ and $\beta = 0$ represents the infinite temperature thermal state. We note that the mapping to free fermions allows Pauli operators to be written in terms of Majorana fermion operators which, in turn, allows an exact calculation of the OTOC (Appendix A). These OTOCs are shown in FIG. (1). For the case of $\lambda = 0$, the Hamiltonian $H_0$ is a combination of two decoupled Majorana chains with symmetric butterfly velocities (Appendix A), and we observe that the right and left butterfly velocities are equal to each other despite the lack of inversion symmetry (left panel). For $\lambda = 1$, the Hamiltonian $H_1$ is the well-known Ising model and butterfly velocities are symmetric, as shown in the middle panel of FIG. (1). By contrast, for the general case $\lambda \in (0,1)$, the Hamiltonian does not have inversion symmetry and the OTOCs show asymmetric butterfly velocities (right panel).

The asymmetry in butterfly speeds for $0 < \lambda < 1$ can be directly understood using the quasiparticle description of the free model. It is known that the butterfly speed in an integrable model is the maximum quasiparticle group velocity [66,67,73], and the operator fronts generically broaden either diffusively or sub-diffusively depending on whether the integrable system is interacting or not [73].

The quasi-particle dispersion for the Hamiltonian in Eq. (2) is $\epsilon_{\lambda,1(2)}(q) = J\big[(1-\lambda)(h_{yz}-h_{zy})\sin q + (-)\big((1-\lambda)^2(h_{yz}+h_{zy})^2\sin^2 q + \lambda^2(h_{zz}^2+h_x^2-2h_{zz}h_x\cos q)\big)^{1/2}\big]$. The butterfly speed to the right (left) is the magnitude of the maximal (minimal) quasi-particle group velocity [66,67,73]

$$
v_{B,\lambda}^r = \max_q \frac{d\epsilon_{\lambda,1(2)}(q)}{dq}, \quad v_{B,\lambda}^l = -\min_q \frac{d\epsilon_{\lambda,1(2)}(q)}{dq}. \tag{4}
$$

These are plotted in FIG. (2), where asymmetric butterfly velocities are clearly observed when $\lambda$ is $\in (0,1)$. For the special cases of $\lambda = 0$ and $\lambda = 1$, the right and left butterfly speeds are the same

$$
v_{B,0}^r = v_{B,0}^l = 2J \max(|h_{yz}|, |h_{zy}|), \tag{5}
$$
$$
v_{B,1}^r = v_{B,1}^l = J \min(|h_{zz}|, |h_x|). \tag{6}
$$

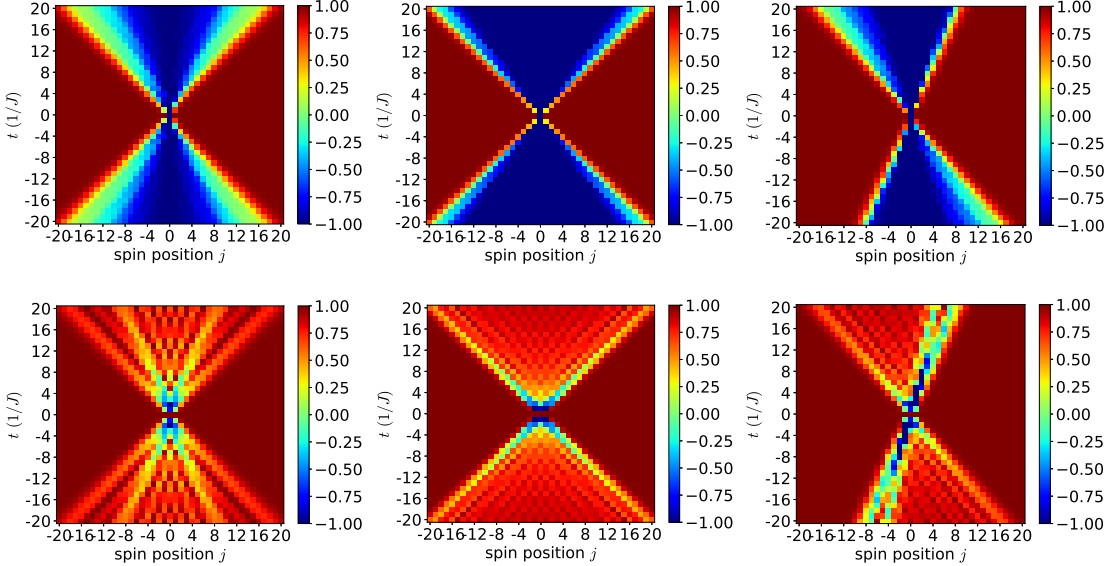

Figure 1: OTOCs $C_{xz}(j,t)$ (upper panel) and $C_{xx}(j,t)$ (lower panel) in the Hamiltonian $H_\lambda$ [Eq. (2)] with parameters $h_{yz} = 0.5, h_{zy} = -0.25, h_{zz} = 1.0, h_x = -1.05$, and $\lambda = 0$ in the left panel, $\lambda = 1$ in the middle panel, and $\lambda = 0.5$ in the right panel. The asymmetric light-cone is clear in the right panel.

The above results are consistent with the butterfly velocities demonstrated via the out-of-time-ordered correlations shown in FIG. (1). For the case of $\lambda = 0$, the Hamiltonian $H_0$ is a combination of two decoupled Majorana chains with symmetric butterfly velocities $2J|h_{yz}|$ and $2J|h_{zy}|$, so the butterfly velocities for $H_0$ is $2J \max(|h_{yz}|, |h_{zy}|)$. For $\lambda = 1$, the butterfly velocities depend on the minimal of $|h_x|$ and $|h_{zz}|$.

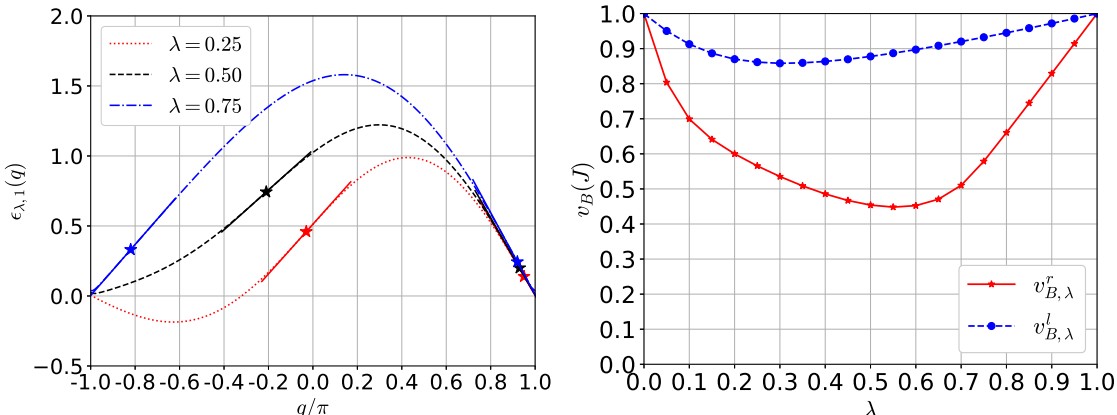

Figure 2: Quasi-particle dispersion relations $\epsilon_{\lambda,1}(q)$ (left panel) and asymmetric butterfly speeds (right panel) for the Hamiltonian $H_\lambda$ [Eq. (2)]. The parameters used are $h_{yz} = 0.5, h_{zy} = -0.25, h_{zz} = 1.0, h_x = -1.05$. In the left panel, the star $\star$ denotes the place where the dispersion relation has maximal or minimal slope, and the solid lines represent the slope. In the right panel, asymmetric butterfly speeds are directly determined from the quasi-particle dispersion relations [Eq. (4)].

# 3 Non-integrable Hamiltonians

In this section, we construct non-integrable Hamiltonian by adding longitudinal fields to the free Hamiltonian $H_\lambda$ [64,65]. The asymmetric butterfly velocities are estimated from a variety of measures including out-of-time-ordered correlations, right/left weight of time-evolved operators, and operator entanglement.

The interacting Hamiltonian on a one dimensional lattice with open boundary conditions is

$$H = \frac{-J(1-\lambda)}{2} \sum_{j=1}^{L-1} \Big[ h_{yz}\sigma_j^y \sigma_{j+1}^z + h_{zy}\sigma_j^z \sigma_{j+1}^y \Big]$$

$$+ \frac{-J\lambda}{2} \Big[ h_{zz} \sum_{j=1}^{L-1} \sigma_j^z \sigma_{j+1}^z + \sum_{j=1}^{L} h_x \sigma_j^x \Big] - \frac{J}{2}\Big[ \sum_{j=1}^{L} h_z \sigma_j^z \Big], \tag{7}$$

where $L$ is the system size, and $h_z$ is a longitudinal field strength. We select the particular parameters $\lambda = 0.5, h_{yz} = 0.5, h_{zy} = -0.25, h_{zz} = 1.0, h_x = -1.05, h_z = 0.5$, although none of our results are fine tuned to this choice.

The longitudinal field breaks integrability and is expected to thermalize the system. For non-integrable Hamiltonians with thermalizing dynamics, the level statistics is consistent with the distribution of level spacings in random matrix ensembles [74]. Let $E_0 < \cdots < E_n < E_{n+1} < \cdots$ be the sequence of ordered energy eigenvalues and $s_n = (E_{n+1} - E_n)$ be the level spacings. One defines the ratio of consecutive level spacings $r_n = s_n/s_{n-1}$, and the distribution of $r_n$ can be described by the Wigner-like surmises for non-integrable systems [75,76]

$$p_W(r) = \frac{1}{Z_W} \frac{(r+r^2)^W}{(1+r+r^2)^{1+3W/2}}, \tag{8}$$

where $W = 1, Z_1 = 8/27$ for Gaussian Orthogonal Ensemble (GOE), and $W = 2$, $Z_2 = 4\pi/(81\sqrt{3})$ for Gaussian Unitary Ensemble (GUE), while they are Poissonian for integrable systems. As shown in FIG. (3), the ratio distribution provides evidence supporting the non-integrability of the Hamiltonian. When $\lambda = 0.5$, the Hamiltonian is complex Hermitian, and its ratio distribution agrees with the Gaussian Unitary Ensemble (GUE). When $\lambda = 1$, the Hamiltonian is real, symmetric and has the inversion symmetry with respect to its center, and its level statistics in the sector with even parity agrees with the Gaussian Orthogonal Ensemble (GOE) [64,65].

We now characterize the asymmetric spreading of quantum information in this model using two complementary methods for computing butterfly speeds that are known in the literature. Both methods agree on the estimation of butterfly speeds within the accuracy of finite-size numerics, and both show a strong asymmetry between $v_B^l$ and $v_B^r$.

## 3.1 Asymmetric butterfly velocities from OTOCs

In this subsection, we estimate the asymmetric butterfly velocities from OTOCs.

As discussed earlier, as the time-evolved operator spreads ballistically, OTOCs can detect the light cone and butterfly velocities. The saturated value of OTOCs equals approximately one outside the ballistic light cone and zero inside it. Near the boundary of the light cone, the OTOCs decay in a universal form $C(j,t) = 1 - f\, e^{-c(j-v_B t)^\alpha/t^{\alpha-1}}$ [66,67], where $c, f$ are constants, $v_B$ describes the speed of operator spreading, and $\alpha$ controls the broadening of the operator fronts. In a generic "strongly quantum" system (i.e. away from large $N$/ semiclassical/weak coupling limits) the operator front shows broadening which corresponds to $\alpha > 1$ so that the OTOC is not a simple exponential in $t$ [67].

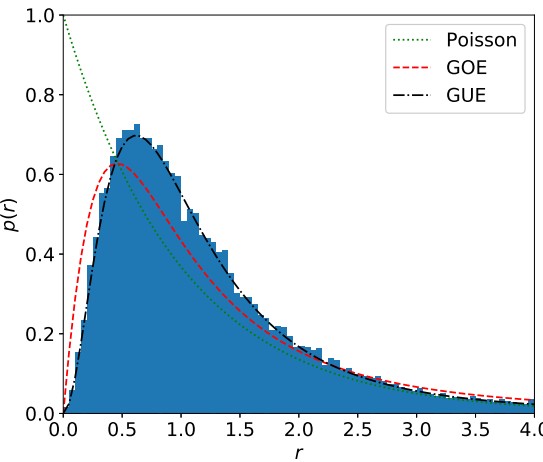

Figure 3: The histogram of the ratio of consecutive level spacings. It is computed from 32768 all energy eigenvalues of the Hamiltonian [Eq. (7)] with parameters $\lambda = 0.5, h_{yz} = 0.5, h_{zy} = -0.25, h_{zz} = 1.0, h_x = -1.05, h_z = 0.5$, and length $L = 15$.

Nevertheless, the decay can still look exponential along rays $j = vt$ in spacetime, $C(j = vt, t) = 1 - f\, e^{\lambda(v)t}$, defining velocity-dependent Lyapunov exponents (VDLEs) which look like $\lambda(v) \sim -c(v - v_B)^\alpha$ near $v_B$ [67]. The VDLEs provide more information about the operator spreading than the butterfly velocities alone.

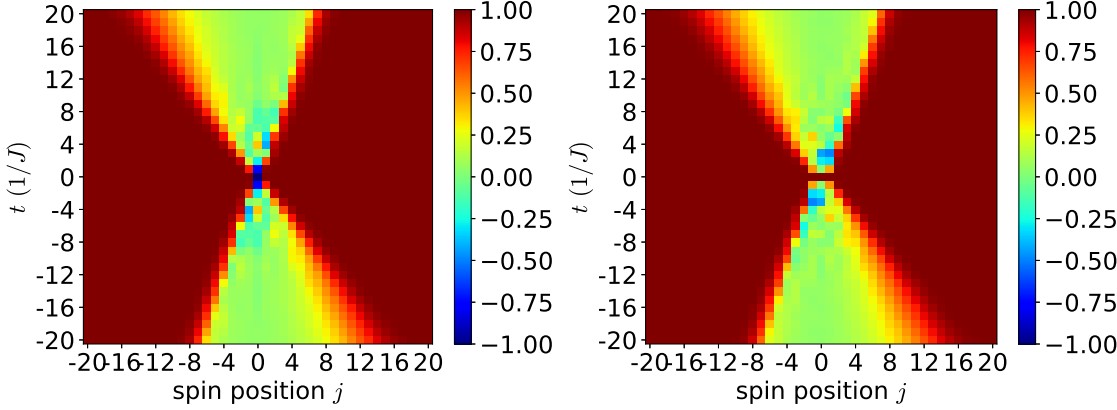

Figure 4: OTOCs $C_{xz}(j, t)$ (left panel) and $C_{xx}(j, t)$ (right panel) in the Hamiltonian $H$ [Eq. (7)] with parameters $\lambda = 0.5, h_{yz} = 0.5, h_{zy} = -0.25, h_{zz} = 1.0, h_x = -1.05, h_z = 0.5$, and length $L = 41$.

First, as shown in FIG. (4), we observe asymmetric butterfly velocities in relatively large systems with $L = 41$ spins. In our numerical calculations, we use the time-evolving block decimation (TEBD) algorithm after mapping matrix product operators to matrix product states [77–79], which is able to efficiently simulate the evolution of operators in the Heisenberg picture. In the numerical simulation, we ignore the singular values $s_k$ if $s_k/s_1 < 10^{-8}$ in the step of singular value decomposition, where $s_1$ is the largest singular value. And the bond dimension is enforced as $\chi \leq 500$. The OTOCs shown in FIG.(4) clearly demonstrate asymmetric butterfly velocities.

Second, we estimate the asymmetric butterfly velocities from the extracted VDLEs $\lambda(v) \sim -c(v - v_B)^\alpha$. Because of the limited computational resources for exact diagonaliza-

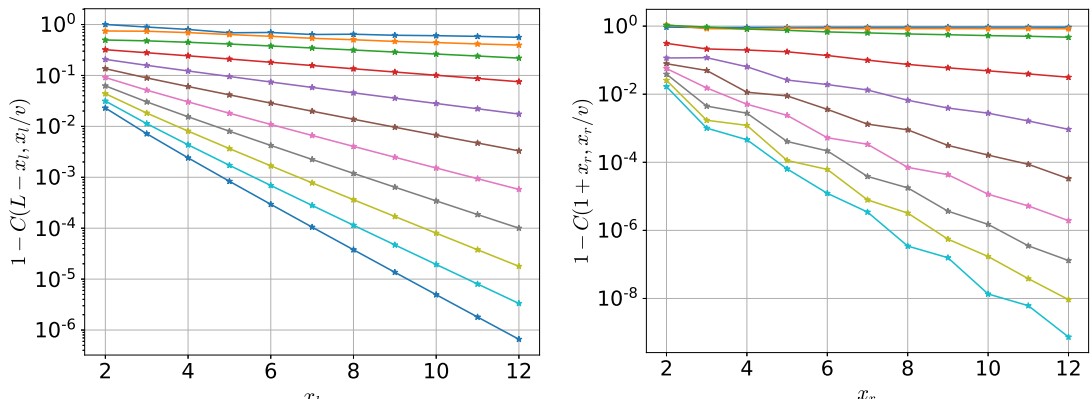

Figure 5: OTOCs in the Hamiltonian $H$ [Eq. (7)] with parameters $\lambda = 0.5, h_{yz} = 0.5, h_{zy} = -0.25, h_{zz} = 1.0, h_x = -1.05, h_z = 0.5$, and length $L = 14$. The left (right) panel shows the left (right) propagating OTOCs along rays at different velocities. Exponential decay can be observed which is consistent with the negative VDLEs for large $v$.

tion, the right and left butterfly velocities are measured by setting the initial local operator at the boundary $j = 1$ and $j = L$ respectively. In FIG. (5), the OTOCs exponentially decay along the rays with different speed $C(1 + x_r, x_r/v) = \langle \sigma_1^x(x_r/v)\sigma_{1+x_r}^z \sigma_1^x(x_r/v)\sigma_{1+x_r}^z \rangle_{\beta=0}$ and $C(L - x_l, x_l/v) = \langle \sigma_L^x(x_l/v)\sigma_{L-x_l}^z \sigma_L^x(x_l/v)\sigma_{L-x_l}^z \rangle_{\beta=0}$. For a given velocity $v$, $\lambda(v)/v$ is the slope of logarithm of the left and right propagating OTOCs versus the distance $x$. After extracting the VDLEs $\lambda(v)$ from the OTOCs, here we give a rough estimation of the butterfly velocities via fitting the curve $\lambda(v) \sim -c(v - v_B)^\alpha$. In FIG. (6), we obtain the results $v_B^r \sim 0.29J$ and $v_B^l \sim 0.66J$.

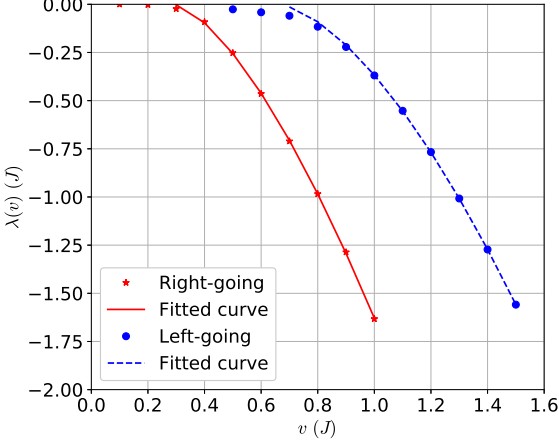

Figure 6: VDLEs fitted from the left and right propagating OTOCs in FIG. (5) with the slope equaling $\lambda(v)/v$. The parameters $c, v_B, \alpha$ can be fitted via the least square method. Here the results of fitting the last 7 points are $v_B^r \sim 0.29J, \alpha^r \sim 1.52$, and $v_B^l \sim 0.66J, \alpha^l \sim 1.61$.

## 3.2 Asymmetric butterfly velocities from right/left weights

Now we turn to the analysis of asymmetric butterfly velocities directly measured from right and left weights of the spatial spreading operators.

To define the right/left weight, note that every operator in a spin 1/2 system with length $L$ can be written in the complete orthogonal basis of $4^L$ Pauli strings $S = \otimes_{j=1}^{L} S_j$, i.e. $O(t) = \sum_S a_S(t)S$, where $S_j = I, \sigma^x, \sigma^y$ or $\sigma^z$. Unitary evolution preserves the norm of operators, so $\sum_S |a_S(t)|^2 = 1$ holds for a normalized operator. The information of operator spreading is contained in the coefficients $a_S(t)$. In order to describe the spatial spreading, the right weight is defined by

$$\rho_r(j,t) = \sum_{S:S_j \neq I, S_{j'>j}=I} |a_S(t)|^2, \tag{9}$$

where the left weight is defined analogously. Because of the conservation of operator norm $\sum_j \rho_{r(l)}(j,t) = 1$, the weight can be interpreted as an emergent local conserved density for the right/left fronts of the spreading operator.

Recent studies [58–61] showed that the hydrodynamics for the right/left weight can be characterized by a biased diffusion equation in non-integrable systems, which means that the front is ballistically propagating with diffusively broadening width. Thus, when the time-evolved operator spreads, $\rho_r$ moves to the right with velocity $v_B^r$, and $\rho_l$ moves to the left with velocity $v_B^l$.

Here in the numerical calculations of exact diagnolization, the right and left weights are obtained by setting the initial local operator at the boundary $j = 1$ and $j = L$ respectively. The right weight $\rho_r(1+x_r,t)$ of $\sigma_1^x(t)$ is calculated in order to compare the left weight $\rho_l(L-x_l,t)$ of $\sigma_L^x(t)$, where $x_r(x_l)$ is the distance between the right (left) end and the location of initial operator. As shown in FIG. (7), the estimated velocities are $v_B^r \sim 0.30J$ and $v_B^l \sim 0.65J$ by fitting the times when the weights reach the maximum peak for given distances. This is in very good agreement with the values obtained from OTOCs in the prior subsection, especially considering the finite resolution of our methods given the limited system sizes and times accessible to numerics.

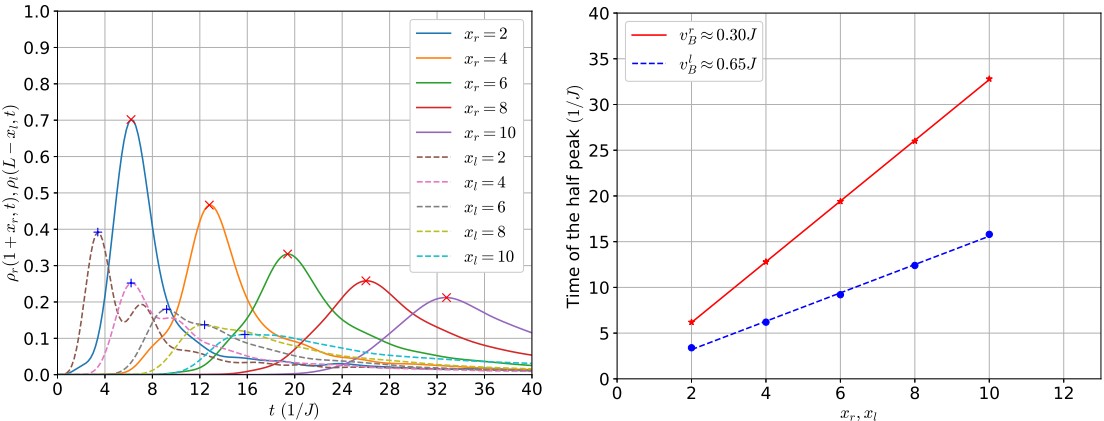

Figure 7: Left panel: the right weights (solid lines) of $\sigma_1^x(t)$ and the left weights (dashed lines) of $\sigma_L^x(t)$. The parameters in the Hamiltonian $H$ [Eq. (7)] are $\lambda = 0.5, h_{yz} = 0.5, h_{zy} = -0.25, h_{zz} = 1.0, h_x = -1.05, h_z = 0.5$, and length $L = 14$. The symbols ×/+ mark the times when the right/left weights reach the maximum peak for given distances. Right panel: time of the peak versus the distance. The solid and dashed lines are the results of linear fitting.

# 4  Conclusion and Discussion

In summary, we have constructed a physically transparent family of integrable Hamiltonians with asymmetric information spreading, and shown that this asymmetric transport persists even upon adding interactions. Exact solutions of the butterfly velocities are obtained in the integrable models while, in the non-integrable case, the asymmetric butterfly velocities are numerically estimated from different quantities characterizing operator spreading including out-of-time-ordered correlations and right/left weight of time-evolved operators. Our constructions present simple mechanisms for obtaining asymmetric transport in simple free-fermion models and spin chains, without invoking notions such as anyonic particle statistics that were previously thought to necessary for asymmetric transport [69].

Given the constructions and studies in this paper, several open questions would be interesting to explore in the future work. Here we have focused on the information spreading at infinite temperature in one dimension. How does the asymmetric spreading change at finite temperature, or in higher dimensional systems? Additionally, it is worth studying how asymmetries encoded in various quantities are intertwined with each other. For example, does the transport of conserved quantities (like energy) inherit the same signatures of asymmetry as the spreading of local operators? Is it possible to disentangle them? Our strategy of starting with free models also seems promising for answering these more general conceptual questions. For example, these ideas could be explored in higher dimensions by constructing free models without radially symmetric dispersions.

Finally, probing the asymmetry of information propagation may be also interesting to explore in many-body localized systems or disordered systems with Griffiths effects, where the butterfly velocities are zero and the light cones are logarithmic or sub-ballistic.

# Acknowledgements

We thank Xie Chen, Shenghan Jiang, Kevin Slage, and Cheng-Ju Lin for helpful discussions. VK thanks Charles Stahl and David Huse for collaboration on related work. Y.-L.Z. is supported by the National Science Foundation under Award Number DMR-1654340, and the Institute for Quantum Information and Matter, an NSF Physics Frontiers Center (NSF Grant PHY-1733907).

# Appendix A: Analytic solution of time-evolved operators and OTOCs in the free model

The Jordan-Wigner mapping allows spin operators to be written in terms of free Majorana fermions as follows: : $\sigma_j^x = i\gamma_{2j}\gamma_{2j+1}$, $\sigma_j^z = (\prod_{k=-\infty}^{j-1} i\gamma_{2k}\gamma_{2k+1})\gamma_{2j}$ and $\sigma_j^y = (\prod_{k=-\infty}^{j-1} i\gamma_{2k}\gamma_{2k+1})\gamma_{2j+1}$. Then the Hamiltonian [Eq. (2), FIG. (8)] is

$$H_\lambda = (1-\lambda)\frac{-J}{2}\sum_j \left[h_{yz}(-i\gamma_{2j}\gamma_{2j+2}) + h_{zy}(i\gamma_{2j+1}\gamma_{2j+3})\right]$$

$$+ \lambda\frac{-J}{2}\sum_j \left[h_{zz}(i\gamma_{2j+1}\gamma_{2j+2}) + h_x(i\gamma_{2j}\gamma_{2j+1})\right]. \tag{10}$$

Below, we obtain analytic solutions for time-evolved operator for this Hamiltonian [Eq. (2)] within the Heisenberg picture. Denoting $\gamma_0(t) = \sum_n f_n(t)\gamma_n$ and $\gamma_1(t) = \sum_m h_m(t)\gamma_m$, the time-evolved operator is $\sigma_0^x(t) = i\gamma_0(t)\gamma_1(t) = i\sum_{n<m} F_{n,m}(t)\gamma_n\gamma_m$, where

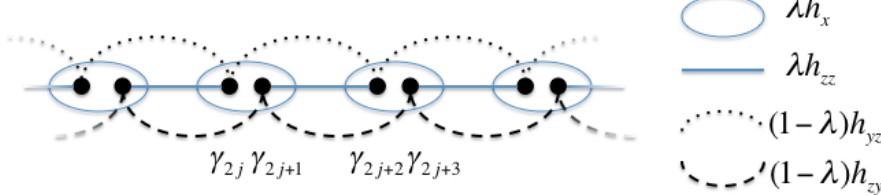

Figure 8: Majorana fermion representation for the Hamiltonian [Eq. (2)]: the solid points represent Majorana fermions, the ellipses represent the local field at each site, and the lines connecting Majorana fermions denote the nearest or next-nearest neighobor hopping terms.

$F_{n,m}(t) = f_n(t)h_m(t) - f_m(t)h_n(t)$, and the out-of-time-ordered correlations are

$$C_{xz}(j,t) = 1 - 2 \sum_{n \leq 2j, m \geq 2j+1} |F_{n,m}(t)|^2, \tag{11}$$

$$C_{xx}(j,t) = 1 - 2\Big[ \sum_{n < 2j} \big(|F_{n,2j}(t)|^2 + |F_{n,2j+1}(t)|^2\big)$$

$$+ \sum_{m > 2j+1} \big(|F_{2j,m}(t)|^2 + |F_{2j+1,m}(t)|^2\big)\Big]. \tag{12}$$

Next, we get the analytic solution of time-evolved operators $\gamma_0(t) = \sum_n f_n(t)\gamma_n$ and $\gamma_1(t) = \sum_m h_m(t)\gamma_m$ in the integrable Hamiltonian [Eq. (2)]. Plugging the candidate solution into the Heisenberg equation, it is straightforward to get the differential equations for the coefficients $f_n(t)$

$$\begin{cases} \frac{df_{2n}(t)}{dt} &= -\lambda J h_{zz} f_{2n-1}(t) + \lambda J h_x f_{2n+1}(t) \\ &\quad + (1-\lambda)J h_{yz}[-f_{2n+2}(t) + f_{2n-2}(t)], \\ \frac{df_{2n+1}(t)}{dt} &= -\lambda J h_x f_{2n}(t) + \lambda J h_{zz} f_{2n+2}(t) \\ &\quad + (1-\lambda)J h_{zy}[-f_{2n-1}(t) + f_{2n+3}(t)], \end{cases}$$

where the initial condition is $f_n(0) = \delta_{n,0}$. After applying the Fourier transformation $f_{2n}(t) = \frac{1}{2\pi} \int_{-\pi}^{\pi} dq\, e^{-inq} A(q,t), f_{2n+1}(t) = \frac{1}{2\pi} \int_{-\pi}^{\pi} dq\, e^{-inq} B(q,t)$, we get

$$\begin{cases} \frac{\partial A(q,t)}{\partial t} &= \lambda J[h_x - h_{zz}e^{iq}]B(q,t) + (1-\lambda)2iJh_{yz}\sin(q)A(q,t), \\ \frac{\partial B(q,t)}{\partial t} &= \lambda J[-h_x + h_{zz}e^{-iq}]A(q,t) - (1-\lambda)2iJh_{zy}\sin(q)B(q,t). \end{cases} \tag{13}$$

Then the analytic solution is

$$\begin{cases} f_{2n}(t) &= \frac{1}{2\pi} \int_{-\pi}^{\pi} dq\, e^{-inq} \frac{\epsilon_{\lambda,1}e^{i\epsilon_{\lambda,1}t} - \epsilon_{\lambda,2}e^{i\epsilon_{\lambda,2}t}}{\epsilon_{\lambda,1} - \epsilon_{\lambda,2}}, \\ f_{2n+1}(t) &= \frac{1}{2\pi} \int_{-\pi}^{\pi} dq\, e^{-inq} \lambda J[-h_x + h_{zz}e^{-iq}] \times \frac{(-i)(e^{i\epsilon_{\lambda,1}t} - e^{i\epsilon_{\lambda,2}t})}{\epsilon_{\lambda,1} - \epsilon_{\lambda,2}}, \end{cases} \tag{14}$$

where
$$\epsilon_{\lambda,1(2)}(q) = J\Big[(1-\lambda)(h_{yz}-h_{zy})\sin q + (-)\big((1-\lambda)^2(h_{yz}+h_{zy})^2\sin^2 q + \lambda^2(h_{zz}^2 + h_x^2 - 2h_{zz}h_x\cos q)^{1/2}\big)\Big].$$

Similarly the coefficients in the exact solution $\gamma_1(t) = \sum_m h_m(t)\gamma_m$ are

$$\begin{cases} h_{2m}(t) &= \frac{1}{2\pi} \int_{-\pi}^{\pi} dq\, e^{-imq} \lambda J[h_x - h_{zz}e^{iq}] \times \frac{(-i)(e^{i\epsilon_{\lambda,1}t} - e^{i\epsilon_{\lambda,2}t})}{\epsilon_{\lambda,1} - \epsilon_{\lambda,2}}, \\ h_{2m+1}(t) &= \frac{1}{2\pi} \int_{-\pi}^{\pi} dq\, e^{-imq} \frac{\epsilon_{\lambda,1}e^{i\epsilon_{\lambda,2}t} - \epsilon_{\lambda,2}e^{i\epsilon_{\lambda,2}t}}{\epsilon_{\lambda,1} - \epsilon_{\lambda,2}}. \end{cases} \tag{15}$$

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
