# Peer review of "Asymmetric butterfly velocities in 2-local Hamiltonians"

_SciPost Physics, doi:SciPost Phys. 9, 024 (2020)_

## Round 1 · Referee Report · Anonymous (Referee 1) · 2020-2-2

Strengths
1- Construction of a simple solvable model with asymmetric light cones 2- Comparison of various numerical methods 3- Clear presentation
Weaknesses
1- Lack of a more careful assessment of the strengths/weaknesses of each method 2- Somewhat limited scope of the results
Report
The authors of this manuscript study the transport of quantum information, as characterized by the spreading of local operators, in one-dimensional systems where inversion symmetry is explicitly broken. In this case, operator spreading can be asymmetric between left and right, characterized by two different 'butterfly' velocities.. They first construct a model that is exactly solvable by a Jordan-Wigner mapping to free fermions, where the two velocities can be analytically calculated, and are shown to be different generically. They then go on to numerically study a non-integrable extension of the model, extracting both velocities using three separate methods (velocity-dependent Lyapunov exponents, operator weights and operator entanglement).
The paper is clearly written and the results presented are sound, albeit of somewhat limited interest - asymmetric light cones were studied previously in the literature, as the authors note in their introduction. I believe it would eventually be suitable for publication in SciPost, but only after some modifications and extensions. In particular, the three numerical methods used to extract v_l,r give significantly different results, making it difficult to discern what the actual butterfly velocities are. Since the comparison of these various methods is a major part of the paper, a more careful assessment of their particular merits seems necessary. In particular, since the authors present a solvable model, it would be quite natural to benchmark all three methods on this test case, to evaluate the accuracy of each. I believe, doing so would significantly strengthen the paper.
Some smaller comments: - I think it would be useful to write explicitly the real-space free fermionic version of the Hamiltonian Eq. (2) in the appendix, it would help with developing some intuition for this model. - The bond dimensions used for TEBD simulations should be stated somewhere. Especially, since the maximal times obtained (t=20) are in fact rather large, and it seems like at this point there is already a quite significant amount of operator entanglement (as shown by Fig. 8) - In their evaluation of v_l,r from operator weights, presented in Fig 7, it is unclear to me why the authors use the half of the peak, rather than the peak itself, as their reference point. Since the wavefront broadens in time, as they note in the introduction, it seems like this could affect the velocity they extract? - In the caption of Fig. 1, when stating the parameters of the model used, say "h_z = 0.5". I believe this is a typo, as these results are supposed to come from the integrable case, which has h_z = 0 (indeed, the Hamiltonian Eq. (2) refered to in the same caption does not have h_z among its parameters) - In the 3rd sentence of the caption of Fig. 2, the first plot is referred to as the 'upper' panel: it should be 'left'. Also, in the same figure, I would suggest making the stars that denote the points corresponding to v_l,r larger to be more visible. - Some typos: "spacial and "one can get the growth function is" (should be "as", presumably).
Requested changes
1- Test methods on solvable model 2- Provide more detail on TEBD simulations (e.g. bond dimensions used, truncation errors, etc) 3- Explain why half peak is used instead of the peak in Fig. 7
Author: Yongliang Zhang on 2020-07-13 [id 880]
(in reply to Report 1 on 2020-02-02)
We that the referee for their positive assessment and their insightful and helpful comments. Below is our reply to the comments and the changes made accordingly.
1. “… - asymmetric light cones were studied previously in the literature, as the authors note in their introduction. … In particular, the three numerical methods used to extract v_l,r give significantly different results, making it difficult to discern what the actual butterfly velocities are. Since the comparison of these various methods is a major part of the paper, a more careful assessment of their particular merits seems necessary. In particular, since the authors present a solvable model, it would be quite natural to benchmark all three methods on this test case, to evaluate the accuracy of each. I believe, doing so would significantly strengthen the paper.”
Our central result is the construction of physically transparent and solvable models that demonstrate asymmetric information spreading, and their generalizations to non-integrable cases. These models and mechanisms are orthogonal to prior constructions in the literature, and they provide a counterexample to prior claims that the anyonic statistics are required for asymmetric information spreading.
In contrast, the different methods presented for computing butterfly speeds were simply meant to be illustrative, since all these methods have been discussed previously in the literature and all clearly show unequal left/right speeds. However, we take the referee’s point that the large amount of space devoted to these methods overemphasizes their importance to this paper and perhaps distracts from our main message. To address this, we have:
- Slightly edited the introduction and conclusion to more clearly state our main results.
- Removed the last method for computing butterfly speeds using operator entanglement, which was subject to the largest amount of numerical uncertainty and did not add any qualitatively new insight.
- Following the referee’s suggestion (point 4 below), we changed our method of estimating speeds from left/right weights to fit to the peak rather than the half-peak, which is indeed less affected by front-broadening effects. As a result of this change, the speeds obtained using the OTOCs and the right/left weights now agree much more closely, ameliorating the referee’s concerns that these methods give different results.
- Finally, while the benchmarking of the methods against free systems would be nice, in practice the operator dynamics in free models has a lot of additional structure (for example, showing revivals) so that some of our methods are not well suited to free systems. Hence a detailed benchmarking of the kind the referee suggests is challenging to perform. However, as mentioned above, our modified analysis of left/right weights already gives much better agreement between methods, largely addressing the referee’s concern about discerning the actual butterfly speeds.
2. “I think it would be useful to write explicitly the real-space free fermionic version of the Hamiltonian Eq. (2) in the appendix, it would help with developing some intuition for this model.”
Thank you for this constructive suggestion. We have now explicitly written the real-space free-fermionic version of the Hamiltonian Eq. (2) in the appendix. In addition, we have plotted a figure to demonstrate the nearest and next-nearest neighbor hopping terms.
3. “The bond dimensions used for TEBD simulations should be stated somewhere. Especially, since the maximal times obtained (t=20) are in fact rather large, and it seems like at this point there is already a quite significant amount of operator entanglement (as shown by Fig. 8).”
Under Fig. 4, we have added a sentence to provide this information. The maximum bond dimension used is 500 and note that log(500) ~ 6.2 is approximately able to handle the operator entanglement at time t=20 (as shown by Fig. 8).
4. “In their evaluation of v_l,r from operator weights, presented in Fig 7, it is unclear to me why the authors use the half of the peak, rather than the peak itself, as their reference point. Since the wavefront broadens in time, as they note in the introduction, it seems like this could affect the velocity they extract?”
Thank you for raising this question. We have changed our figures and the fitting results using the peaks because of the broadening of the wavefront. This also gives much better agreement between the velocities obtained using the OTOCS and left-right operator weights, partially addressing the first remark.
5. “In the caption of Fig. 1, when stating the parameters of the model used, say "h_z = 0.5". I believe this is a typo, as these results are supposed to come from the integrable case, which has h_z = 0 (indeed, the Hamiltonian Eq. (2) refered to in the same caption does not have h_z among its parameters)”
We have removed “h_z=0.5” and corrected this.
6. “In the 3rd sentence of the caption of Fig. 2, the first plot is referred to as the 'upper' panel: it should be 'left'. Also, in the same figure, I would suggest making the stars that denote the points corresponding to v_l,r larger to be more visible.”
Thank you for your careful reading. We have corrected the typos in the caption of Fig. 2 and enlarged the stars to increase visibility.
7. “Some typos: "spacial and "one can get the growth function is" (should be "as", presumably).”
Thanks, we have fixed the typos.
Author: Yongliang Zhang on 2020-07-13 [id 881]
(in reply to Report 2 on 2020-02-13)We thank the referee for their positive assessment and constructive suggestions. Here is our reply to the comments and the changes made accordingly.
1. “A subjective comment is — at times the paper reads like a follow up of past works of some of the authors. May be a more self-contained discussion of the various (borrowed) aspects might improve the readability of this paper.”
We have now modified the introduction and conclusion to better highlight the main message of the paper. Also, as discussed in the reply to Referee 1, we have streamlined and shortened the various methods of analysis presented for computing butterfly speeds, which will hopefully also help with readability.
2. “lambda = 0, 1 case : For the lambda=0 case the vB is symmetric (Fig 1) despite the lack of inversion symmetry, is it obvious why?”
When \lambda = 0, the Hamiltonian H_0 is a combination of two decoupled Majorana chains with symmetric butterfly velocities. We have added sentences to explain this point below Eq. (3) and (5)(6), explicitly written the real-space free-fermionic version of the Hamiltonian Eq. (2), and added a figure representing the Majorana hopping terms in Appendix A.
3. “It might be interesting to understand the role (and physics) of the constants h_ij or h_k in controlling the butterfly velocity.”
Thanks for raising this interesting question. For the integrable Hamiltonian,
(i) \lambda = 0: H_0 is a combination of two decoupled Majorana chains with symmetric butterfly velocities with 2 J |h_{yz}| and 2J |h_{zy}|, so the butterfly velocities for H_0 is 2 J max(|h_{yz}|, |h_{zy}|).
(ii) \lambda = 1: the butterfly velocities depend on the minimal of |h_x| and |h_{zz}|.
(iii) \lambda in (0, 1): the formula of exact solution is complicated, it is hard to discuss the controlling effect of each parameter.
The (weakly) non-integrable Hamiltonian follows these trends roughly, although the relations are no longer exact.
4. “Could the authors briefly comment on whether any of the features they discuss are in any sense restricted to the (a) dimensionality (1D), or (b) zero temperature solutions of the problem.”
(a) In this manuscript, we focus on the asymmetric light cone in 1D systems where the asymmetry can be quantified by the right and left butterfly velocities. Indeed generalizations to higher dimensions is an interesting direction for future study, and we don’t expect the phenomenon to be limited to 1D. For instance, our strategy of starting with free models seems promising for exploring asymmetric information scrambling in higher dimensional systems by constructing free models without radially symmetric dispersions.
(b) The butterfly velocities we discussed are with respect to the infinite temperature \beta = 0, and these velocities are expected to depend on temperature in general. However, we do not expect that the asymmetric butterfly velocities will change to be symmetric when the temperature changes from \beta = 0 to \beta != 0. Again the free model helps provide intuition for this – the butterfly speeds in the free model are derived from quasiparticle velocities, and the temperature simply controls the occupation of quasiparticles but not destroy the asymmetry between left/right moving speeds which is a result of the asymmetric dispersion relation.
5. “There are few small typos that I came across: ‘Jordan Wigner’ after eq. 2 has missing hyphen (also in App. A); ‘about center’ in the same paragraph, probably has missing article; Figure 2 uses ‘upper panel’ which is absent; In the conclusion ‘may also interesting..’ has missing verb; same paragraph has ‘subbalistic’, missing hyphen.”
Thank you for your careful reading. We have fixed the typos.

---

## Round 1 · Referee Report · Anonymous (Referee 2) · 2020-2-13

Strengths
1) Well written 2) Relevant topic
Weaknesses
1) Results are a bit elementary
Report
In this work, the authors demonstrate the asymmetry in the speed of local operator spreading (or butterfly velocity, vB) in a particular 1D theory. By mapping this theory to a free Hamiltonian they compute the OTOC, hence vB, exactly. Using three different complementary methods they obtain similar conclusions in a related non-integrable theory.
The paper is clearly written and presents original results. Therefore, I recommend the paper for publication after the following comments are addressed:
1) A subjective comment is — at times the paper reads like a follow up of past works of some of the authors. May be a more self-contained discussion of the various (borrowed) aspects might improve the readability of this paper.
2) lambda = 0, 1 case : For the lambda=0 case the vB is symmetric (Fig 1) despite the lack of inversion symmetry, is it obvious why?
3) It might be interesting to understand the role (and physics) of the constants h_ij or h_k in controlling the butterfly velocity.
4) Could the authors briefly comment on whether any of the features they discuss are in any sense restricted to the (a) dimensionality (1D), or (b) zero temperature solutions of the problem.
Requested changes
There are few small typos that I came across: ‘Jordan Wigner’ after eq. 2 has missing hyphen (also in App. A); ‘about center’ in the same paragraph, probably has missing article; Figure 2 uses ‘upper panel’ which is absent; In the conclusion ‘may also interesting..’ has missing verb; same paragraph has ‘subbalistic’, missing hyphen.

---

## Round 4 · Referee Report · Anonymous (Referee 2) · 2020-7-13

Report

The authors have adequately addressed all my comments.

---

## Round 4 · Referee Report · Anonymous (Referee 1) · 2020-7-22

Report

The authors have addressed all the questions raised in my previous report. In particular, I am glad to see that the two methods of estimating the butterfly velocities now agree to a good precision.

---

## Round 4 · Author Response

Dear Editor,

Thank you and the referees for reviewing our manuscript "Asymmetric butterfly velocities in 2-local Hamiltonians" (scipost_201912_00045v1). We are grateful for the referees’ comments and suggestions to improve our manuscript. We have replied to the referee reports using SciPost comments, and would like to resubmit our improved manuscript to SciPost Physics as a regular article. Thank you very much for your consideration.

Sincerely yours,
Yong-Liang Zhang and Vedika Khemani

---

## Round 4 · List of Changes

(1) We have slightly edited the introduction and conclusion to better emphasize the main results of the work and how it adds to previous results on asymmetric speeds in the literature, including correcting the misconception that asymmetric speeds require anyonic statistics.

(2) We previously presented three different methods for computing butterfly speeds, leading the first referee to conclude that this was a central part of our paper. In fact, these different methods are not essential for illustrating our central result of presenting simple and solvable models with asymmetric butterfly speeds, and we agree with the referee that the large amount of space devoted to the different methods distracts from our central message. To address this, we have removed the last method which was subject to the largest amount of numerical uncertainty and did not add any qualitatively new insight.

(3) In the caption of Fig. (1), we have removed “h_z=0.5”. Also, we have fixed the typos that the referees pointed out. In Fig. (2), we have enlarged the stars to increase visibility.

(4) Below Eq. (3) and (5)(6), we have added sentences to explain that for the lambda=0 case, the v_B is symmetric despite the lack of inversion symmetry.

(5) Under Fig. (4), we have provided parameters for TEBD simulations.

(6) In Fig. (7), we have changed our figures and the fitting results using the peaks because of the broadening of the wavefront. Corresponding estimation results have been changed, and show better agreement between the two methods of computing butterfly speeds.

(7) In Appendix A, we have explicitly written the real-space free-fermionic version of the Hamiltonian Eq. (2). In addition, we have plotted a figure to demonstrate the nearest and next-nearest neighbor hopping terms.

---

## Editorial Decision

published